# Delayed blood pressure recovery after exercise stress test is associated with autonomic dysfunction and pulse pressure in a middle-aged healthy group

Hancheol Lee[�����], Hyeongsoo Kim[�����], Seungjin Oh, Jong-Kwan Park, Ji-Yong Jang, Kyeong-Hyeon Chun, Se-Jung Yoon[ID]*

Division of Cardiology, National Health Insurance Service Ilsan Hospital, Goyang, Republic of Korea

These authors contributed equally to this work.
* drpuoohsjy@gmail.com

## Abstract

### Background

Delayed heart rate (HR) and blood pressure recovery after exercise test is known as the reliable indexes of autonomic dysfunction. Here we tried to evaluate the serial changes in various indicators during exercise test and correlations with recovery of HR and blood pressure in a normotensive healthy middle-aged group.

### Methods

A total of 122 patients without hypertension or diabetes was enrolled (mean age, 55.6 ± 11.0; male, 56.6%; mean blood pressure, 124.8 ± 16.6 / 81.5 ± 9.6 mmHg). Treadmill test was performed for evaluation of chest pain. Patients with coronary artery disease, positive treadmill test result, left ventricular dysfunction or renal failure were excluded. Heart rate recovery was calculated by subtracting the HR in the first or second minute of recovery period from the HR of peak exercise (HRR1 or HRR2). Systolic blood pressure in the $4^{th}$ minute of recovery stage (SBPR4) was used to show delayed blood pressure recovery.

### Results

Metabolic equivalents (METs) and HR in stage 2 to 4 were significantly correlated with both HRR1 and HRR2. Multiple regression analysis of HRR revealed significant correlation of METs and SBPR4. SBPR4 was significantly correlated with both HRR1 and HRR2 (HRR1, r = -0.376, p<0.001; HRR2, r = -0.244, p = 0.008) as well as SBP in the baseline to stage 3 and pulse pressure (r = 0.406, p<0.001).

### Conclusions

Delayed BP recovery after peak exercise test revealed significant association with autonomic dysfunction and increased pulse pressure in normotensive middle-aged healthy

**Data Availability Statement:** The data contain sensitive patient information. Researchers may send data access requests to Ju-Hee Ahn, Medical

Laboratory Technologist, [ajh1441@nhimc.or.kr] who is directly conducting treadmill tests of all patients, and organizing and managing data related to the test at the authors' institution.

**Funding:** This study was funded by NATIONAL HEALTH INSURANCE SERVICE ILSANHOSPITAL (NHIMC2017CR078). The funders had no role in study design, data collection and analysis, decision to publish, or preparation of the manuscript.

**Competing interests:** The authors have declared that no competing interests exist.

group. It can be a simple and useful marker of autonomic dysfunction and arterial stiffness.

## 1. Introduction

Exercise stress test is easy to perform and useful physiological test. It has the advantage of being able to check the serial changes of vital signs during gradual loading.

Delayed heart rate recovery (HRR) was well known as the difference between the heart rate of peak exercise and a specific stage during the recovery time in a patient undergoing a maximal stress test. It has been revealed to be associated with the balance of sympathetic and parasympathetic tonus. HRR is a reflection of vagal reactivation and impaired HRR is considered to indicate a blunted reactivation of vagal tone [1–3].

HRR abnormalities are often seen in patients with metabolic disorders, including cardiovascular disease, fatty liver, diabetes and prehypertension [4–8]. Blunted HRR has been reported the association with chronic heart failure and new-onset atrial fibrillation [9, 10]. It has been revealed as an independent predictor of mortality and adverse outcomes [11–18]. Mahfouz *et al.* [19], suggested that abnormal HRR and delayed systolic blood pressure recovery after exercise were correlated with impaired endothelial function and diastolic dysfunction in prediabetics.

Systolic blood pressure at recovery stage (SBPR) immediately after exercise also has been found to have diagnostic value for coronary artery disease and myocardial perfusion abnormalities [20, 21]. Although not as well known as HRR, it was also known to be affected by autonomic function [22, 23] and has been known to have a strong relationship with cardiovascular disease and overall mortality [23–27]. To the best of our knowledge, no study has been found on the correlation between the hemodynamic indices and autonomic nerve function at each stage in the exercise stress test. Moreover, association of SBPR with HRR and other serial exercise indices has not been identified in peak exercise test of middle-aged healthy non-hypertensives. Here we tried to evaluate the serial changes in various indicators and correlations among them including HRR and SBPR of peak exercise test in a normotensive healthy middle-aged group.

## 2. Materials and methods

### 2.1. Study participants

The study population consisted of individuals referred for treadmill exercise test for the evaluation of chest pain between January 2014 and December 2017. Patients under 18 years of age, severe obesity (BMI $\geq$ 35 Kg/m$^2$), positive treadmill test result, medical history of hypertension, diabetes mellitus, dyslipidemia, any cardiovascular disease, left ventricular dysfunction, valvular heart disease, atrial fibrillation or renal failure were excluded.

### 2.2. Protocol of the exercise stress test

The patients underwent a standard maximal graded exercise treadmill test according to the standard Bruce protocol with a T2100-ST2 Treadmill system (GE Inc., Boston, MA). Continuous 12-lead electrocardiographic monitoring was performed throughout testing. The Tango exercise BP monitoring device (SunTech Medical, Morrisville, NC) was used to automatically measure each subject's BP and HR before and at the second minute of each stage of the

exercise. The participants exercised until the HR achieved was >95% of estimated maximal HR (220 −age). The patients continued to walk for 30 seconds at a speed of 1.5 mph during the recovery period, after which they sat down with continued BP and HR

Monitoring [19, 28]. HRR values were calculated by subtracting the HR at the first, second and fourth minute of the recovery period from the HR reached at peak exercise. The exercise capacity was calculated as total metabolic equivalent units (METs) achieved at peak exercise.

### 2.3. Statistical analysis

All analyses were made using the SPSS 20.0 for Windows software package (SPSS Inc., Chicago, IL, USA). Continuous variables were presented as a mean ± standard deviation and categorical variables as a percentage of the group total. Pearson's correlation analyses were performed to determine the association of HRR and SBP in the 4th minute of recovery stage (SBPR4) with other indicators of exercise test. A stepwise, multiple regression analysis was used to identify significant determinants of HRR in the first and second minute of recovery stage, which included variables that correlated with a P-value of less than 0.1 in the Pearson's correlation analysis. A P-value of less than 0.05 was considered significant.

## 3. Results

### 3.1. Baseline characteristics and exercise data of treadmill test

A total of 122 patients without diabetes or hypertension were enrolled (mean age, 55.6 ± 11.0; male, 56.6%; mean baseline blood pressure, 124.8 ± 16.6 / 81.5 ± 9.6 mmHg). Mean body mass index (BMI) was under obesity (25.0 ± 4.6 Kg/m$^2$) and exercise capacity is satisfactory (11.6 ± 1.9 metabolic equivalent (METs)) (Table 1).

### 3.2. Correlation of HRR with other hemodynamic index

Baseline BP, baseline HR, peak BP were not significantly correlated with HRR in the first and second minute of recovery stage (HRR1 and HRR2). However age, metabolic equivalents (METs), HR in stage 2 to 4 and SBPR4 were significantly correlated with both HRR1 and HRR2 (HRR1, r = 0.248, p = 0.006; HRR2, r = 0.308, p = 0.001 with HR at stage 2) (Table 2). Multiple regression analysis of HRR1 and HRR2 revealed significant correlation with METs and SBPR4 (Table 3, Fig 1).

### 3.3. Correlation of BP at recovery stage with other hemodynamic index including HRR

SBPR4 revealed significant correlation with SBP at baseline, stage 1~3 each (r = 0.537, p<0.001 at baseline; r = 0.595, p<0.001 at stage 1; r = 0.575, p<0.001 at stage 2; r = 0.567, p<0.001 at stage 3). HR at recovery stage (1st and 2nd minute) (r = 0.205, p = 0.029 at 1st minute of recovery stage; r = 0.193, p = 0.040 at 2nd minute of recovery) and HRR (r = -0.376, p<0.001 HRR1; r = -0.244, p = 0.008 HRR2) were significantly correlated with SBPR4.

Pulse pressure was significantly correlated with SBPR4 (r = 0.406, p<0.001). On the contrary, DBP in the 4th minute of recovery stage showed significant correlation only with DBP at each stage (Table 4, Fig 2).

## 4. Discussion

In this study, we have performed the analysis in a middle-aged healthy group without medical history of hypertension or diabetes mellitus, who had good exercise capacity (mean exercise capacity of 11.6 and mean age of 55.6). We investigated variables associated with HRR, an

**Table 1. Baseline characteristics and exercise data of treadmill test.**

| | Total (N = 122) mean ± SD, N (%) |
|---|---|
| Age | 55.6 ± 11.1 |
| Sex (male) | 69 (56.6) |
| BMI (Kg/m$^2$) | 25.0 ± 4.6 |
| Exercise capacity (METs) | 11.6 ± 1.9 |
| Exercise Time (sec) | 606.2 ± 112.6 |
| Baseline | |
| SBP at baseline (mmHg) | 124.8 ± 16.6 |
| DBP at baseline (mmHg) | 75.5 ± 9.6 |
| HR at baseline (bpm) | 75.9 ± 11.4 |
| Stage 1 | |
| SBP at stage 1 (mmHg) | 134.4 ± 21.3 |
| DBP at stage 1 (mmHg) | 76.3 ± 10.1 |
| HR at stage 1 (bpm) | 104.6 ± 14.5 |
| Stage 2 | |
| SBP at stage 2 (mmHg) | 142.6 ± 23.7 |
| DBP at stage 2 (mmHg) | 75.4 ± 10.3 |
| HR at stage 2 (bpm) | 121.8 ± 17.0 |
| Stage 3 | |
| SBP at stage 3 (mmHg) | 149.7 ± 27.1 |
| DBP at stage 3 (mmHg) | 79.1 ± 13.2 |
| HR at stage 3 (bpm) | 142.9 ± 19.0 |
| Stage 4 | |
| SBP at stage 4 (mmHg) | 155.6 ± 30.4 |
| DBP at stage 4 (mmHg) | 81.4 ± 19.2 |
| HR at stage 4 (bpm) | 164.4 ± 19.1 |
| Recovery stage | |
| SBP in the 4th minute (mmHg) | 143.5 ± 21.4 |
| DBP in the 4th minute (mmHg) | 79.1 ± 12.0 |
| HR in the 1st minute (bpm) | 128.3 ± 20.3 |
| HR in the 2nd minute (bpm) | 107.6 ± 19.5 |
| HR in the 4th minute (bpm) | 95.8 ± 17.1 |
| HRR1 (bpm) | 30.6 ± 12.1 |
| HRR2 (bpm) | 51.4 ± 17.2 |
| HRR4 (bpm) | 64.2 ± 17.2 |
| Pulse pressure (mmHg) | 43.3 ± 15.1 |
| Mean baPWV (cm/sec) | 1429.2 ± 249.1 |
| hfPWV (cm/sec) | 953.8 ± 250.1 |

Data are presented as mean ± SD or number of patients (%).
Abbreviations. BMI, body mass index; METs, metabolic equivalents; SBP, systolic blood pressure; DBP, diastolic blood pressure; HR, heart rate; HRR1,2,4, heart rate recovery in the first, second and forth minute of recovery stage; baPWV, ankle-brachial pulse wave velocity; hfPWV, heart-femoral pulse wave velocity.

indicator of autonomic nerve function, during normal exercise load tests and found that systolic blood pressure at recovery stage was significantly associated. And systolic blood pressure at recovery stage showed significant relevance to SBP at each stage, HRR and pulse pressure in middle-aged healthy group.

**Table 2. Correlation of HRR with other hemodynamic index.**

| | HRR in the 1st minute of recovery stage | | HRR in the 2nd minute of recovery stage | |
|---|---|---|---|---|
| | Pearson Correlation | p-value | Pearson Correlation | p-value |
| age | -0.206 | 0.023* | -0.159 | 0.082 |
| BMI | 0.376 | 0.093 | 0.352 | 0.117 |
| METs | 0.181 | 0.046* | 0.234 | 0.010* |
| Exercise time | 0.046 | 0.618 | 0.145 | 0.117 |
| Baseline | | | | |
| SBP at baseline | -0.080 | 0.384 | -0.048 | 0.061 |
| DBP at baseline | 0.076 | 0.408 | 0.600 | 0.510 |
| HR at baseline | 0.068 | 0.456 | 0.091 | 0.322 |
| Stage 1 | | | | |
| SBP at stage 1 | -0.121 | 0.195 | -0.146 | 0.116 |
| DBP at stage 1 | 0.101 | 0.281 | 0.100 | 0.281 |
| HR at stage 1 | 0.150 | 0.103 | 0.167 | 0.070 |
| Stage 2 | | | | |
| SBP at stage 2 | -0.063 | 0.501 | -0.084 | 0.365 |
| DBP at stage 2 | 0.126 | 0.175 | 0.096 | 0.300 |
| HR at stage 2 | 0.248 | 0.006* | 0.308 | 0.001* |
| Stage 3 | | | | |
| SBP at stage 3 | -0.079 | 0.443 | -0.086 | 0.404 |
| DBP at stage 3 | 0.291 | 0.004* | 0.296 | 0.003* |
| HR at stage 3 | 0.281 | 0.002* | 0.403 | <0.001* |
| Stage 4 | | | | |
| SBP at stage 4 | 0.028 | 0.869 | -0.215 | 0.202 |
| DBP at stage 4 | 0.354 | 0.031* | 0.475 | 0.003* |
| HR at stage 4 | 0.279 | 0.006* | 0.399 | <0.001* |
| Recovery stage | | | | |
| HR in the 1st minute | -0.112 | 0.220 | 0.130 | 0.155 |
| HR in the 2nd minute | -0.181 | 0.047* | -0.264 | 0.003* |
| HR in the 4th minute | -0.188 | 0.042* | -0.117 | 0.207 |
| HRR1 | - | - | 0.778 | <0.001* |
| HRR2 | 0.778 | <0.001* | - | - |
| HRR4 | 0.732 | <0.001* | 0.806 | <0.001* |
| SBPR4 | -0.376 | <0.001* | -0.244 | 0.008* |
| DBPR4 | 0.018 | 0.848 | 0.052 | 0.582 |
| Pulse pressure | -0.136 | 0.138 | -0.091 | 0.321 |
| Mean baPWV | -0.056 | 0.559 | -0.133 | 0.163 |
| HfPWV | -0.014 | 0.891 | -0.117 | 0.252 |

Abbreviations. BMI, body mass index; METs, metabolic equivalents; SBP, systolic blood pressure; DBP, diastolic blood pressure; HR, heart rate; HRR1,2,4, heart rate recovery in the first, second and forth minute of recovery stage; SBPR4, SBP in the 4th minute of recovery stage; DBPR4, DBP in the 4th minute of recovery stage; baPWV, ankle-brachial pulse wave velocity; hfPWV, heart-femoral pulse wave velocity.

*$p < 0.05$

Based on these results, HRR gets blunted down as aging and decreased exercise intensity. The increase in heart rate during exercise was significantly proportional to HRR and rapid decrease in heart rate during recovery showed significant negative correlation with HRR in this group. This indicates that HR, which rises high during exercise and rapidly decreases during recovery appears as desirable HRR and can represent good autonomic function.

**Table 3. Correlation of HRR with other basic and hemodynamic parameters.**

| HRR 1 | | | | | |
|---|---|---|---|---|---|
| **Model 1** | **Unstandardized Regression Coefficient** | | **Standardized Regression Coefficient** | **t** | **p-value** |
| **variable** | **B** | **standard error** | **β** | | |
| (Constant) | 45.767 | 14.512 | | 3.154 | 0.002 |
| Age | -0.154 | 0.103 | -0.140 | -1.487 | 0.140 |
| BMI | 0.035 | 0.235 | 0.014 | 0.149 | 0.882 |
| **METs** | 2.092 | 0.983 | 0.295 | 2.128 | 0.036* |
| Exercise Time | -0.016 | 0.014 | -0.151 | -1.134 | 0.259 |
| **SBP at stage 1** | 0.125 | 0.062 | 0.233 | 2.031 | 0.045* |
| **SBPR4** | -0.272 | 0.065 | -0.488 | -4.192 | <0.001* |
| HRR 2 | | | | | |
| Model 1 | Unstandardized Regression Coefficient | | Standardized Regression Coefficient | t | p-value |
| variable | B | standard error | β | | |
| (Constant) | 32.856 | 20.256 | | 1.622 | 0.108 |
| age | -0.058 | 0.142 | -0.040 | -0.411 | 0.682 |
| BMI | 0.410 | 0.326 | 0.119 | 1.257 | 0.211 |
| **METs** | 3.004 | 1.371 | 0.313 | 2.191 | 0.031* |
| Exercise Time | 0.000 | 0.019 | 0.003 | 0.021 | 0.983 |
| SBP at stage 1 | 0.094 | 0.085 | 0.128 | 1.100 | 0.274 |
| **SBPR4** | -0.249 | 0.087 | -0.338 | -2.873 | 0.005* |

*$p < 0.05$

dependent variables: HRR1min and HRR 2min

Abbreviations. HRR1,2, heart rate recovery in the first and second minute of recovery stage; BMI, body mass index; METs, metabolic equivalents; SBP, systolic blood pressure; SBPR4, SBP in the 4th minute of recovery stage.

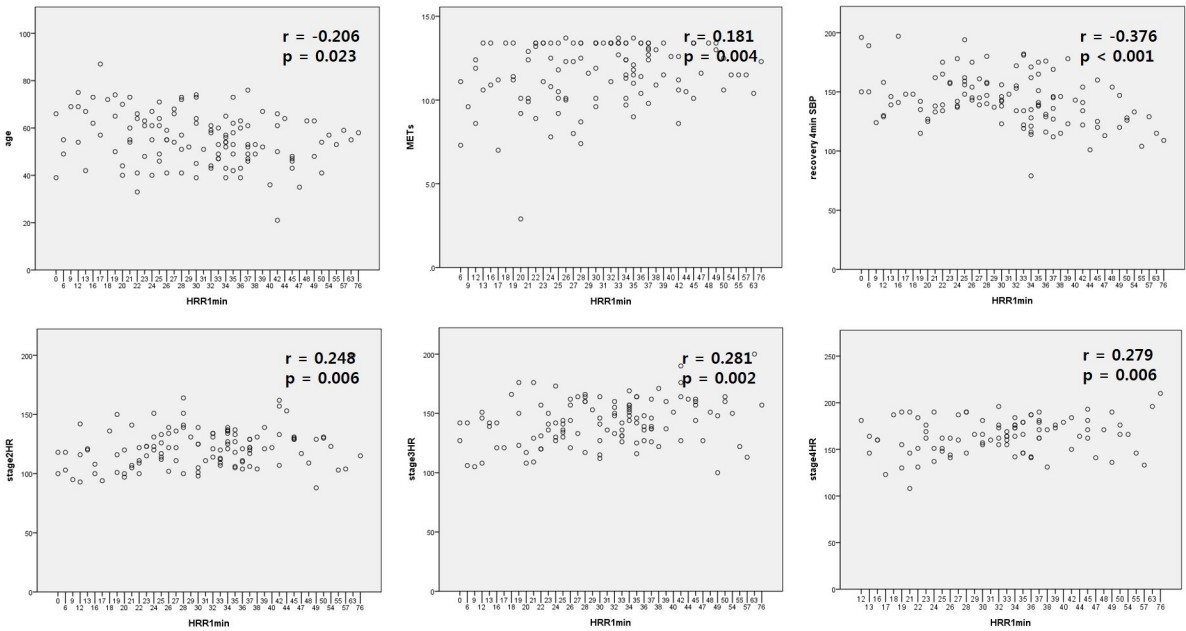

**Fig 1. Correlation of HRR in the first minute of recovery stage with other basic and hemodynamic parameters.** Abbreviations. HRR1min, HRR in the first minute of recovery stage, METs, metabolic equivalents; recovery4min SBP, systolic blood pressure in the 4th minute of recovery stage; HR, heart rate. *$p < 0.05$.

**Table 4. Correlation of blood pressure in the 4th minute of recovery stage with other hemodynamic indexes.**

| | SBP in the 4th minute of recovery stage | | DBP in the 4th minute of recovery stage | |
|---|---|---|---|---|
| | Pearson Correlation | p-value | Pearson Correlation | p-value |
| Age | 0.088 | 0.347 | -0.109 | 0.246 |
| BMI | 0.074 | 0.753 | -0.115 | 0.628 |
| Exercise capacity (METs) | 0.074 | 0.430 | 0.079 | 0.405 |
| Exercise time | 0.052 | 0.580 | 0.167 | 0.076 |
| Baseline | | | | |
| SBP at baseline | 0.537 | <0.001* | 0.184 | 0.051 |
| DBP at baseline | 0.286 | 0.002* | 0.555 | <0.001* |
| HR at baseline | 0.165 | 0.079 | 0.125 | 0.186 |
| Stage 1 | | | | |
| SBP at stage 1 | 0.595 | <0.001* | 0.219 | 0.020* |
| DBP at stage 1 | 0.233 | 0.013* | 0.577 | <0.001* |
| HR at stage 1 | -0.072 | 0.447 | -0.061 | 0.519 |
| Stage 2 | | | | |
| SBP at stage 2 | 0.575 | <0.001* | 0.081 | 0.390 |
| DBP at stage 2 | 0.184 | 0.050 | 0.493 | <0.001* |
| HR at stage 2 | -0.118 | 0.213 | -0.056 | 0.556 |
| Stage 3 | | | | |
| SBP at stage 3 | 0.567 | <0.001* | 0.127 | 0.223 |
| DBP at stage 3 | 0.067 | 0.541 | 0.380 | <0.001* |
| HR at stage 3 | -0.085 | 0.375 | -0.100 | 0.296 |
| Stage 4 | | | | |
| SBP at stage 4 | 0.285 | 0.097 | 0.088 | 0.617 |
| DBP at stage 4 | 0.017 | 0.922 | 0.296 | 0.084 |
| HR at stage 4 | -0.028 | 0.790 | 0.101 | 0.337 |
| Recovery stage | | | | |
| HR in the 1st minute | 0.205 | 0.029* | 0.032 | 0.735 |
| HR in the 2nd minute | 0.193 | 0.040* | -0.024 | 0.799 |
| HR in the 4th minute | 0.145 | 0.124 | 0.041 | 0.667 |
| HRR1 | -0.376 | <0.001* | -0.018 | 0.848 |
| HRR2 | -0.244 | 0.008* | 0.052 | 0.582 |
| HRR4 | -0.170 | 0.070 | -0.016 | 0.867 |
| SBP in the 4th minute of recovery stage | | | 0.374 | <0.001* |
| DBP in the 4th minute of recovery stage | 0.374 | <0.001* | | |
| Pulse pressure | 0.406 | <0.001* | -0.158 | 0.093 |
| Mean baPWV | 0.112 | 0.250 | 0.186 | 0.073 |
| hfPWV | 0.034 | 0.745 | 0.037 | 0.726 |

Abbreviations. BMI, body mass index; METs, metabolic equivalent; SBP, systolic blood pressure; DBP, diastolic blood pressure; HR, heart rate; HRR1,2,4, heart rate recovery in the first, second and forth minute of recovery stage; baPWV, ankle-brachial pulse wave velocity; hfPWV, heart-femoral pulse wave velocity.

*p < 0.05

We can also see that the better the HRR, the faster the SBPR stabilizes. It seems that the better the autonomic nerve functions, the faster the blood pressure and HR after exercise are stabilized due to harmonious reactivation of parasympathetic tone after peak exercise.

SBPR is analyzed to be significantly related to SBP before and during exercise. Additionally, arterial stiffness represented by pulse pressure showed a significant proportional relationship with SBPR in this study.

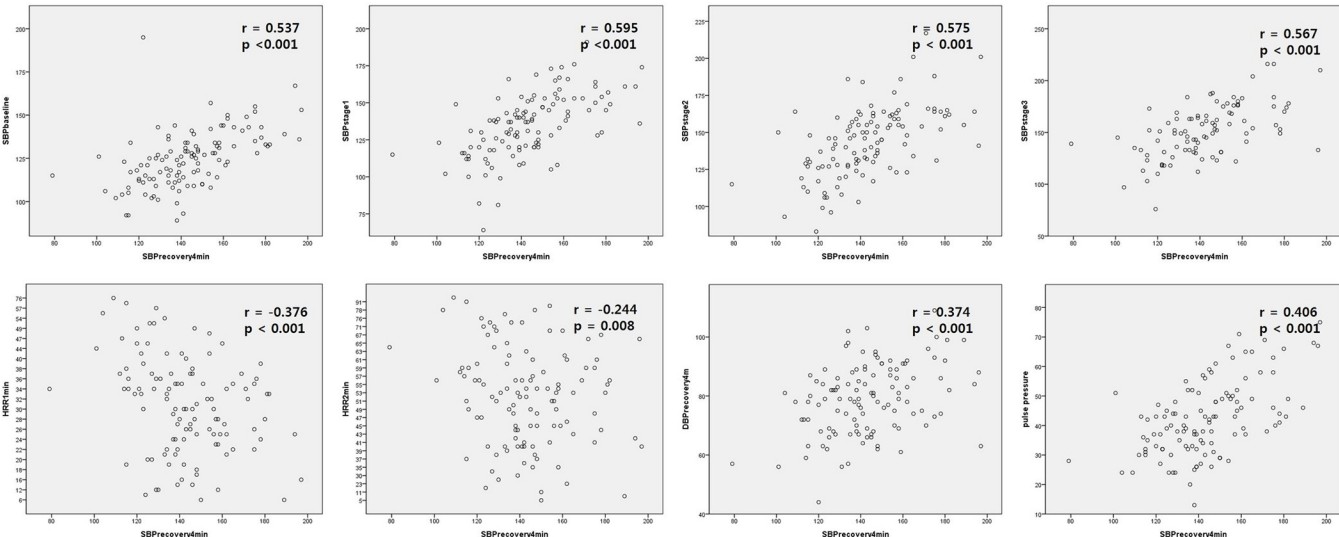

**Fig 2. Correlation of SBP in the 4th minute of recovery stage with other basic and hemodynamic parameters.** Abbreviations. SBPrecovery4min, systolic blood pressure in the 4th minute of recovery stage; DBPrecovery4min, diastolic blood pressure in the 4th minute of recovery stage; HRR1,2min, HRR in the first and second minute of recovery stage. *p < 0.05.

After exercise, HR and BP normally decrease to resting levels via reactivation of vagal tone and withdrawal of sympathetic neural drive in healthy persons [29]. The response of systolic blood pressure to exercise is influenced by two neurohormonal mechanisms. The first involves the parasympathetic and sympathetic efferent changes that result in a cardiovascular response to exercise. The other includes the autonomic efferent response due to intramuscular afferent receptors that are sensitive to the metabolites produced by skeletal muscle [23]. Arteriolar tone is also influenced by the release of these local factors, which include nitric oxide, adenosine, lactate, and the subsequent decline in pH associated with exercise [22].

Schwartz *et al.* have reported that increased vagal tone is associated with improved survival, emphasizing the importance of HRR as a prognostic marker [30]. SBPR immediately after exercise also has been found to have diagnostic value for coronary artery disease and myocardial perfusion abnormalities [20, 21, 25, 26] and it has revealed strong relationship with HRR, cardiac diastolic function, cardiovascular disease and overall mortality including sudden cardiac death [3, 19, 23–27, 29, 31, 32].

A previous study showed similar results with ours. The positive correlation between HRR1 and decrease of systolic blood pressure at the first minute of the recovery stage in subjects without exaggerated blood pressure response to exercise (EBPR) was presented and the enrolled patients had similar characteristics with our study group without medical history of hypertension, diabetes mellitus, cardiovascular or other systemic disease [33].

In another study, SBPR1 and 2 values were observed to be significantly blunted in the metabolic syndrome group. They thought that autonomic and endothelial dysfunction, which has previously been well established in patients with metabolic syndrome, might play an essential role in these impaired SBPR values [3]. Kontsas *et al.* presented that delayed blood pressure response detected during recovery stage implied a reverse relationship with peripheral vascular resistance during exercise in treated hypertensives using pulse wave velocity (PWV) and blood pressure recovery ratio [29]. Recently significant association between HRR1 and augmentation index was reported, which demonstrated the correlation of HRR with arterial stiffness. In this data, we did not show significant correlation between HRR and pulse pressure, but between SBPR4 and pulse pressure [34].

SBPR4 has also been used as an important index in previous study that showed notable difference between two groups of HRR cutoff point of 18 beats per minute. However it was not revealed as a significant predictor of sudden cardiac death and cardiovascular mortality during a follow-up period of 47±13 months [35].

The SBPR decreases after exercise may be a reflection of a person's level of physical activity and fitness. The more rapid decline indicates the higher level of physical fitness, and a greater decrease in SBP from peak exercise to the recovery may reflect good aerobic capacity [16, 23, 27]. Researches also showed age differences in SBPR after exercise with faster recovery of SBP in younger adults than older group [36].

To our knowledge, this is the first study to analyze several continuous and stepwise aerobic exercise indexes and find mutual correlation in the middle-aged healthy group including HRR, SBP in the recovery stage and arterial stiffness with pulse pressure. Delayed SBP recovery after peak exercise test revealed significant association with autonomic dysfunction and increased pulse pressure in this group. It can be a simple and useful marker of autonomic dysfunction and arterial stiffness in the middle-aged healthy group.

There were several limiting factors in our study. Although the research population has relatively homogeneous characteristics, this study is a small sized retrospective, single-center study and needs to be verified in various diseases and conditions. It can be better to be validated by larger, prospective, multicenter studies. Further limitations include a lack of analysis of cardiovascular events or occurrence of metabolic disorders through continuous tracking. Additionally, coronary artery disease is excluded but the patients were not performed any imaging study such as angiography or computed tomography but only treadmill test.

If a large-scale long-term follow-up study in patients with diabetes or metabolic syndrome can be conducted in the future, various promising analyses can be expected.

## 5. Conclusions

In summary, delayed SBP recovery after peak exercise test revealed significant association with reduced HRR and increased pulse pressure in this group. It can be a simple and useful marker of autonomic dysfunction and arterial stiffness in the middle-aged healthy group.

## Author Contributions

**Conceptualization:** Ji-Yong Jang, Se-Jung Yoon.

**Data curation:** Hancheol Lee, Hyeongsoo Kim, Ji-Yong Jang.

**Formal analysis:** Hancheol Lee, Hyeongsoo Kim.

**Funding acquisition:** Hyeongsoo Kim.

**Investigation:** Hancheol Lee, Hyeongsoo Kim, Se-Jung Yoon.

**Methodology:** Hancheol Lee, Hyeongsoo Kim, Ji-Yong Jang.

**Project administration:** Hancheol Lee.

**Resources:** Hancheol Lee, Jong-Kwan Park, Kyeong-Hyeon Chun.

**Software:** Seungjin Oh.

**Supervision:** Jong-Kwan Park, Se-Jung Yoon.

**Validation:** Seungjin Oh, Kyeong-Hyeon Chun, Se-Jung Yoon.

**Visualization:** Seungjin Oh.

**Writing – original draft:** Hancheol Lee.

**Writing – review & editing:** Hancheol Lee, Se-Jung Yoon.

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
