## [Decision Letter · Decision Letter 0]

2 Aug 2023

PONE-D-23-13242Delayed blood pressure recovery after exercise stress test is associated with autonomic dysfunction and pulse pressure in a middle-aged healthy groupPLOS ONE

Dear Dr. Yoon,

Thank you for submitting your manuscript to PLOS ONE. After careful consideration, we feel that it has merit but does not fully meet PLOS ONE’s publication criteria as it currently stands. Therefore, we invite you to submit a revised version of the manuscript that addresses the points raised during the review process.

We look forward to receiving your revised manuscript.

Kind regards,

Damon Leo Swift

Academic Editor

PLOS ONE

Journal Requirements:

- 10.1097/MD.0000000000000428

- https://doi.org/10.3109/08037051.2012.759694

- https://doi.org/10.1016/j.ahj.2003.08.009

In your revision ensure you cite all your sources (including your own works), and quote or rephrase any duplicated text outside the methods section. Further consideration is dependent on these concerns being addressed.

"no conflict of interest to declare."

"There is no conflict of interest."

Additional Editor Comments:

Please edit the present manuscript based on the reviewers comments.

Reviewers' comments:

Reviewer's Responses to Questions

**Comments to the Author**

1. Is the manuscript technically sound, and do the data support the conclusions?

Reviewer #1: Yes

Reviewer #2: Partly

2. Has the statistical analysis been performed appropriately and rigorously? 

Reviewer #1: Yes

Reviewer #2: Yes

3. Have the authors made all data underlying the findings in their manuscript fully available?

Reviewer #1: Yes

Reviewer #2: Yes

4. Is the manuscript presented in an intelligible fashion and written in standard English?

Reviewer #1: Yes

Reviewer #2: No

5. Review Comments to the Author

Reviewer #1: The paper is solid , but the authors could make a stronger case for its importance , as if one is doing a stress test , one already has HRR so why is BP recovery needed . Likewise , why does one need to predict PP when it is so easy to measure by just taking BP?

Reviewer #2: Dear authors,

congratulations for your work about Delayed blood pressure recovery after exercise stress test is associated with autonomic dysfunction and pulse pressure in a middle-aged healthy group. Your aim is not well defined in the manuscript, it is hard for the reader to understand your goal. You must define with more specification your aim and strenght the need for this study in the literature. "Why is important to do this study?"

Suggestions:

1. Abstract: add sentence with background; Results: rewrite this part and remove statistical procedures;

2: Introduction: Add more setences about your topic and exercise type and population; Define with more specificity your study aim;

3: Methods: Add yoiur study design; Add exclusion criteria; Why not perform anova?; Add procedures

4: Results: redone statistical procedures;

5: Discussion: You need to add more specific literature to debate your results. E.g.:"The increase in heart rate during exercise was proportional to HRR in this group. It can be interpreted that HRR will be good if the HR increases sufficiently much during exercise." what is sufficiently much during exercise?

6. PLOS authors have the option to publish the peer review history of their article (what does this mean?). If published, this will include your full peer review and any attached files.

Reviewer #1: No

Reviewer #2: **Yes: **Luis Leitão

---

## [Author Response · Author response to Decision Letter 0]

18 Aug 2023

Rebuttal letter 

We appreciate you for all your kind and high-quality advice and respected and referred to the opinions of editors and reviewers as much as possible. We modified and reviewed them (below).

All revised parts were highlighted and marked on the manuscript.

Journal Requirements:

-We added the funding information to the revised manuscript as you recommended. 

-We modified the manuscript as below according to PLOS ONE's style requirements (TITLE, AUTHOR, AFFILIATIONS FORMATTING GUIDELINES & MANUSCRIPT BODY FORMATTING GUIDELINES). 

1) We removed titles attached to the author's name (Do not include titles (Dr., PhD, M.D,,) 

2) We listed corresponding author’s initials in parentheses after the email address.

3) We removed physical addresses of corresponding author but only mentioned email addresses.

4) We did not include ZIP or Postal Codes, street addresses, or building/office numbers of corresponding author as PLOS ONE's style requirements (Formatting Guidelines)

5) We modified all level 1 headings to be bold and 18pt font and level 2 headings to be bold and 16pt font each. 

6) References ; The authors' names were limited to the first six authors, followed by ‘et al.’.

- 10.1097/MD.0000000000000428

- https://doi.org/10.3109/08037051.2012.759694

- https://doi.org/10.1016/j.ahj.2003.08.009

In your revision ensure you cite all your sources (including your own works), and quote or rephrase any duplicated text outside the methods section. Further consideration is dependent on these concerns being addressed.

-The overlapping text you mentioned mainly corresponds to references 23 and 27, so we found these parts in the text and changed them to different expressions as much as possible (highlighted in the text).

-We specified the grant numbers for the award we have received for this study in the ‘Funding Information’ section and also added this point to the ‘cover letter’.

"no conflict of interest to declare."

-We listed the source of funding and the grant numbers in the manuscript and we clearly state that the funders had no role in study design, data collection and analysis, decision to publish, or preparation of the manuscript in cover letter. 

"There is no conflict of interest."

-We clearly state that the authors have declared that no competing interests exist in cover letter as you recommended. 

-We will provide the access to data such as tables and figures in the manuscript.

Reviewers' comments:

Review Comments to the Author

Reviewer #1: The paper is solid , but the authors could make a stronger case for its importance , as if one is doing a stress test , one already has HRR so why is BP recovery needed . Likewise , why does one need to predict PP when it is so easy to measure by just taking BP?

- Thank you for your advice and we can get a lot of data at once by doing an exercise stress test including HRR, serial blood pressure and HR indexes. When evaluating a person's autonomic nerve function, more reliable and objective evidence can be used by using multiple indicators including systolic blood pressure at recovery stage (SBPR) as well as HRR. Although HRR is a well-known indicator, blood pressure that is not easily normalized even after exercise can provide very useful information to suspect autonomic dysfunction. In this study, it is meaningful to show that systolic blood pressure of recovery stage can be a simple indicator that can suspect autonomic nerve abnormalities when blood pressure does not quickly normalize after usual exercise. 

And as mentioned in the text, it showed that rapid stabilization of blood pressure after exercise was related to arterial stiffness, and through this, systolic blood pressure at recovery stage (SBPR) can give a message that it is not irrelevant to various cardiovascular complications related to vascular stiffness. And we added additional other study results on relationship of SBPR and arterial stiffness to the discussion section (reference 29)

Reviewer #2: Dear authors,

congratulations for your work about Delayed blood pressure recovery after exercise stress test is associated with autonomic dysfunction and pulse pressure in a middle-aged healthy group. Your aim is not well defined in the manuscript, it is hard for the reader to understand your goal. You must define with more specification your aim and strenght the need for this study in the literature. "Why is important to do this study?"

- Thank you for your advice and we modified and supplemented the aim of this study more clearly in the abstract and introduction section.

Furthermore, there was a study result showing the relationship between convalescent blood pressure and cardiovascular events, was added to the reference (reference 32).

32. Laukkanen JA, Willeit P, Kurl S, Mäkikallio TH, Savonen K, Ronkainen K, et al. Elevated systolic blood pressure during recovery from exercise and the risk of sudden cardiac death. J Hypertens. 2014 Mar; 32(3):659-666.

Suggestions:

1. Abstract: add sentence with background; Results: rewrite this part and remove statistical procedures;

- Thank you. As you pointed out, the background and results were modified and supplemented to make the purpose and results of the study more clear. 

2: Introduction: Add more setences about your topic and exercise type and population; Define with more specificity your study aim;

- Thank you for your advice and the purpose of the study was added in more detail in the introduction section. We specifically described the exercise type and population in the ‘Materials and Methods’ section.

3: Methods: Add yoiur study design; Add exclusion criteria; Why not perform anova?; Add procedures

- Thank you for your advice and we mentioned the study design in the section of ‘Materials and Methods’ including study population which referred for treadmill exercise test for the evaluation of chest pain and they underwent a standard maximal graded exercise treadmill test according to the standard Bruce protocol with a T2100-ST2 Treadmill system.

-There is the ‘exclusion criteria’ of this study in ‘study participants’ of ‘Materials and Methods’ section and we added ‘patients under 18 years of age to the exclusion criteria’. ; … Patients under 18 years of age, severe obesity (BMI ≥ 35 Kg/m2), positive treadmill test result, medical history of hypertension, diabetes mellitus, dyslipidemia, any cardiovascular disease, left ventricular dysfunction, valvular heart disease, atrial fibrillation or renal failure were excluded.

-Thank you for your advice and we reviewed again all the statistical process of this study as you mentioned. This study did not use student's t test or ANOVA because the cases were not divided into several groups and compared.

As the baseline characteristics of all cases, the value of mean and standard deviation or number (%) of each item were shown in Table 1. Pearson’s correlation analysis (Table 2) and a stepwise, multiple regression analysis (Table 3) corrected for several factors were performed to see the correlation with HRR among the entire data. Furthermore, pearson’s correlation analysis was used to see a reasonable relationship of SBP during the recovery period, which were the focus of this study with other hemodynamic variables such as blood pressure, heart rate, pulse pressure or PWV other than HRR (Table 4). 

4: Results: redone statistical procedures;

-Same as above.

5: Discussion: You need to add more specific literature to debate your results. E.g.:"The increase in heart rate during exercise was proportional to HRR in this group. It can be interpreted that HRR will be good if the HR increases sufficiently much during exercise." what is sufficiently much during exercise?

-Thank you. I agree with your opinion. And for better understanding about the sentence you asked, let me explain with Table 2: HRR showed a significant positive correlation with the HR in the peak exercise stage and a significant negative correlation with the HR in the recovery period (the 1st to 2nd minute). 

This indicates that HR, which rises high during exercise and rapidly decreases during recovery appears as desirable HRR and can represent good autonomic function. We changed this expression you have asked to be more explicit.

---

## [Decision Letter · Decision Letter 1]

18 Sep 2023

Delayed blood pressure recovery after exercise stress test is associated with autonomic dysfunction and pulse pressure in a middle-aged healthy group

PONE-D-23-13242R1

Dear Dr. Yoon,

We’re pleased to inform you that your manuscript has been judged scientifically suitable for publication and will be formally accepted for publication once it meets all outstanding technical requirements.

Kind regards,

Damon Leo Swift

Academic Editor

PLOS ONE

Additional Editor Comments (optional):

The response to reviewers are adequate for publication

Reviewers' comments:

Reviewer's Responses to Questions

**Comments to the Author**

1. If the authors have adequately addressed your comments raised in a previous round of review and you feel that this manuscript is now acceptable for publication, you may indicate that here to bypass the “Comments to the Author” section, enter your conflict of interest statement in the “Confidential to Editor” section, and submit your "Accept" recommendation.

Reviewer #2: All comments have been addressed

2. Is the manuscript technically sound, and do the data support the conclusions?

Reviewer #2: Partly

3. Has the statistical analysis been performed appropriately and rigorously? 

Reviewer #2: Yes

4. Have the authors made all data underlying the findings in their manuscript fully available?

Reviewer #2: Yes

5. Is the manuscript presented in an intelligible fashion and written in standard English?

Reviewer #2: Yes

6. Review Comments to the Author

Reviewer #2: Congratulations for your work about this topic. You have attended my suggestions and the manuscript improved.

7. PLOS authors have the option to publish the peer review history of their article (what does this mean?). If published, this will include your full peer review and any attached files.

Reviewer #2: **Yes: **Luis Leitão

---

## [Editor Report · Acceptance letter]

25 Sep 2023

PONE-D-23-13242R1 

Delayed blood pressure recovery after exercise stress test is associated with autonomic dysfunction and pulse pressure in a middle-aged healthy group 

Dear Dr. Yoon:

I'm pleased to inform you that your manuscript has been deemed suitable for publication in PLOS ONE. Congratulations! Your manuscript is now with our production department. 

Kind regards, 

on behalf of

Dr. Damon Leo Swift 

Academic Editor

PLOS ONE